# Adolescents’ Opinions on COVID-19 Vaccine Hesitancy: Hints toward Enhancing Pandemic Preparedness in the Future

**DOI:** 10.3390/vaccines11050967

**Published:** 2023-05-10

**Authors:** Alessio Muscillo, Gabriele Lombardi, Elena Sestini, Francesca Garbin, Vittoradolfo Tambone, Laura Leondina Campanozzi, Paolo Pin

**Affiliations:** 1Department of Economics and Statistics, University of Siena, 53100 Siena, Italy; 2Department of Statistics, Computer Science, Applications "Giuseppe Parenti", University of Florence, 50134 Florence, Italy; 3Charles River Associates (CRAI), 80802 Munich, Germany; 4Research Unit of Bioethics and Humanities, Campus Bio-Medico University of Rome, 00128 Rome, Italy; 5Bocconi Institute for Data Science and Analytics (BIDSA), Bocconi Univerity, 20146 Milan, Italy

**Keywords:** vaccine hesitancy, COVID-19, adolescents, trust, ethics, pandemic preparedness

## Abstract

To understand and assess vaccine reluctance, it is necessary to evaluate people’s perceptions and grasp potential reasons for generic apprehension. In our analysis, we focus on adolescents’ impressions towards anti-vaxxer behavior. The aim of the study is to figure out students’ opinions about vaccine reluctance, connecting possible explanations that motivate anti-vaxxer decisions with common specific personality traits. We further investigate people’s forecasts concerning the evolution of the pandemic. Between 2021 and 2022, we conducted a randomized survey experiment on a sample of high school individuals (N=395) living in different Italian regions. At that time, the vaccination campaign had already been promoted for nearly one year. From the analysis, it emerges that vaccinated people (92%), especially males, tend to be more pessimistic and attribute a higher level of generic distrust in science to anti-vaxxers. The results show that family background (mother’s education) represents the most influential regressor: individuals coming from less educated families are less prone to attribute generic distrust and distrust of vaccines as principal reasons for vaccine reluctance. Similarly, those who rarely use social media develop a minor tendency to believe in a generic pessimism of anti-vaxxers. However, concerning the future of the pandemic, they are less likely to be optimistic toward vaccines. Overall, our findings shed light on adolescents’ perceptions regarding the factors that influence vaccine hesitancy and highlight the need for targeted communication strategies to improve vaccination rates.

## 1. Introduction

Worldwide, governments have addressed the COVID-19 pandemic by focusing on the development, testing, and implementation of vaccines, as well as promoting massive immunization campaigns. This approach has produced many benefits for society. Several studies have provided evidence of the safety and effectiveness of COVID-19 vaccines, particularly in preventing hospitalization and the illness itself [1,2,3,4]. Vaccines are classified as one of the most valuable public health measures in preventing disease and death from viruses. However, vaccine hesitancy and anti-vaccination movements [5] have emerged as significant problems. Worldwide, surveys have revealed low willingness to get vaccinated, both before and after vaccines became available [6,7,8,9,10].

Vaccine hesitancy—defined as a delay in acceptance or the refusal of vaccines despite the accessibility of vaccination services [11]—is not a new phenomenon [9,12]. It had already been identified among the top 10 threats to global health in 2019 [13], and it further intensified with the new vaccination campaign against the COVID-19 pandemic [14]. Previous studies highlighted the complexity of vaccine hesitancy and recognized three potential elements affecting decision making: contextual influences, individual and group influences, and vaccine-specific issues [10]. Other works emphasized the importance of people’s knowledge about vaccines and the role of online media in spreading information, mostly considering misinformation as one of the most urgent threats [15,16].

We provide an additional contribution to the literature on vaccine hesitancy by analyzing possible reasons for vaccine rejection through a randomized survey experiment. Our alternative approach involves analyzing people’s feelings and perceptions regarding the behavior of others. The goal of our study is to understand what individuals think about those who reject vaccination, particularly in emergency situations such as the COVID-19 pandemic, and the main reasons behind vaccine reluctance. Increasing confidence and promoting positive expectations among young people are essential for long-term effects on health and well-being [17]. In particular, we focused on young individuals’ beliefs concerning vaccine hesitancy, as it is crucial for population immunity [18]. Among other factors, the opinions of teenagers themselves may have significant peer effects on the attitudes of their relatives and acquaintances [19].

The determinants of vaccine hesitancy have been investigated mostly in the adult population over 18 years old, due to the fact that vaccination strategies were initially planned at the global level for this target audience to tackle the pandemic, with priority given to healthcare workers and frail patients [20,21]. Investigating vaccine hesitancy among teenagers is more challenging. First, they tend to underestimate the paramount importance of vaccination due to their young age, driven by ingenuous optimism about potential COVID-19-associated health risks. In addition, the growing occurrence of psychological difficulties and emotional reactions experienced by adolescents during the pandemic contributed to increased refusal and low compliance with vaccine uptake [22,23].

However, not only can adolescents be severely affected by COVID-19, but they can also contribute to a more rapid spread of the epidemic due to their attendance at schools and many social venues [22,24]. For this reason, strategies aimed at effectively fighting health emergencies must necessarily provide specific immunization plans for this group. The few previous studies investigating vaccine hesitancy among adolescents have mostly addressed parental influence as one of the major elements fostering the intention to get vaccinated in young persons [25]. Consistent with previous studies, a mother’s education plays a relevant role not only in actions undertaken [26], but also in shaping individual beliefs. Moreover, vaccine reluctance was found to be positively related to both the family’s socioeconomic and cultural background and the parents’ high levels of social media usage [26]. Nevertheless, adolescents want to have (and often do have) an active role in decision making about their health issues, and therefore, the beliefs and information they learn from different media also come into play in judgments about vaccination [27,28,29].

To the best of our knowledge, this is the first pilot study aimed at assessing Italian adolescents’ opinions about vaccine hesitancy and its motivations. The randomization framework allows us to analyze the effect of two different arguments on affecting people’s beliefs. The first argument aims to highlight the benefits of the vaccination campaign, seen as either a public good or an individual good [30,31]. This choice is driven by the fact that individuals can act according to the logic of free-riding, attempting to avoid the possible side effects of getting vaccinated by benefiting from the vaccination of others [32]. The second argument investigates the effects of information concerning either mild or severe side effects of vaccines. Therefore, this study will be useful to better understand adolescents’ opinions and to increase vaccine uptake among adolescents to prepare for possible future pandemics.

## 2. Materials and Methods

### 2.1. The Survey

The data come from a survey conducted in Italian high schools in the period mid-October 2021 to mid-February 2022, thanks to a collaboration with the no-profit foundation *Fondazione Mondo Digitale* (FMD), and it was approved by the Ethical Committee of the University of Siena (CAREUS). The survey was implemented online with the platform *Qualtrics*. The link was then distributed to the schools that were in contact with FMD. In this sense, we can safely assume that, although schools self-selected to participate in the project, the individual students did not strongly self-select with regards to answering the survey. The survey is implemented in a way such that every individual is randomly assigned to one of the five treatments, which are described in more detail below.

A stylized flow of the survey is described as follows (see also in Figure 1): it starts with an introduction and a consent request; the subject is asked how often they keep up with the news using online social media or newspapers; the subject answers demographic questions concerning their gender, age, type of school attended, region of residence, dimension of their city, and family socioeconomic background as proxied by the father’s and mother’s education; the subject is asked whether they are vaccinated or, if not, whether they intend to get vaccinated, and what were their motivations for getting vaccinated or not; the random treatment is administered; the subject is asked what they think of those who are not vaccinated or do not want to get vaccinated and what they think their motivations are; lastly, the subject is asked how they think the situation will evolve in the short term and in the long term.

### 2.2. The Treatments

The study aims to test the effects of three arguments on influencing adolescents’ opinions towards vaccination. The first argument emphasizes the benefits of the vaccine campaign as a public good, highlighting the importance of global public goods. The COVID-19 pandemic exposed weaknesses in national governments’ abilities to deal with global health crises, underscoring the need for global public goods [33]. Our study will investigate whether people are more influenced by vaccination presented as a public or private good. Stimulating citizens’ engagement through the promotion of a strong reciprocity culture has a beneficial effect on vaccination intentions and the idea that vaccines are a common good [34,35].

Appealing to self-interest rather than the social contract nature of vaccines has been a common messaging strategy to promote COVID-19 vaccine uptake, stimulating individualism and polarization of opinions. This study will also consider an argument for vaccines as private goods [31]. While collectivism is the main driver of reducing vaccine hesitancy, individualism can foster protective behavior for those who trust medicine and institutions and reduce it for those who mistrust them [36,37,38].

Finally, we will investigate the effect of information about mild and severe side effects of vaccines. The reasoning behind this choice is that individuals may avoid getting vaccinated to avoid possible side effects while still benefiting from herd immunity. Emphasizing only the individual benefits of vaccination can foster free-riding, while focusing on the prosocial benefits of herd immunity can promote vaccination [39,40].

The treatment consists in showing every subject a brief text related to the COVID-19 pandemic. In particular, each subject is randomly assigned to one of the following treatments, which are sections of text extracted from official statistics and whose original source is also reported in the survey. The subject was shown the text in quotation (here translated from the original Italian), while the number and brief description of the treatments are only shown here for clarity:T0*Control* group: no treatment, i.e., no text is shown to the subject.T1The vaccine as a *public good*. “Did you know that: according to the Italian Society of Surgery, 400,000 operations were missed in 2020 due to the Coronavirus and that according to the Italian Federation of Medical Doctors in 2020 there were about 30,000 more deaths for “neglected diseases”, i.e., compared to those attributed to Covid and those expected from other pathologies.”T2The vaccine as a *private good*. “Did you know that: estimates of July 2021 from the ISS (Italian National Institute of Health) say that the percentage of Covid cases among the vaccinated is far lower than the percentage of cases among the unvaccinated and that the vaccine prevents very effectively hospitalization and death. For example, the effectiveness in preventing hospitalization is 94.6% and in preventing death is 95.8%.”T3Some statistics about *severe adverse effects* of the vaccine. “Did you know that: AIFA (Italian Medicines Agency) reports 0.12% of suspicious events after the administration of an anti-Covid vaccine. 13% of these reports concern serious and potentially fatal events.”T4Statistics about *mild adverse effects* of the vaccine. “Did you know that: Pfizer/BioNTech and AstraZeneca vaccines in their information leaflets declare the incidence of mild side effects (such as fever and nausea) in more than one in 10 people. Data on British citizens instead report the appearance of these symptoms in one in 4 people.”

In line with the classification described by Benin and colleagues [41] (accepter, hesitant, late, rejecter), the ideas behind the development of these treatments are the following: T1 and T2 are meant to capture the motivation of the accepters (and possibly the hesitants), be it the fact that the entire society benefits from mass vaccination or be it that a vaccinated individual is less exposed to severe health consequences in case of infection. Indeed, T1 is meant to highlight the societal benefits from mass vaccination and T2 is meant to highlight the advantages accrued to the single individual after vaccination. So, we can expect that both these treatments would incentivize people to get vaccinated and to think negatively of those who choose not to get vaccinated.

Treatments T3 and T4 are designed to capture the motivations of late and rejecter individuals because, by describing possible adverse effects, they can discourage vaccination while encouraging free-riding behaviors. Moreover, by distinguishing between severe (T3) and mild (T4) adverse effects, we are able to capture and control for two different intensities.

As usually performed in randomized controlled trials, the effect of the treatment is then tested on the variables (in our case questions) administered after the treatment itself by comparing the answers of the treated subjects with the answers provided by the control group (T0).

### 2.3. Description of the Sample

The descriptive statistics of the data analyzed are summarized in Table 1, including the information about the principal component analysis described in detail in Section 2.5. The demographics are summarized as follows: two-thirds of the subjects are female, their ages are equally spread from 12 to 19 years, and their mothers’ education is also equally spread between women with no high school degree, with a high school degree, and with a university degree. It is worth noting that, as is usually performed in the literature, not only can parental education be used as a proxy for the familial background, but also the father’s education is less informative than the mother’s (which is also confirmed in our analysis when selecting the variables to be used). The schools attended are mainly classified as *lyceum*, while the rest are schools that, according to the Italian system, are classified as technical or vocational. In terms of city dimension, about 65% live in towns with more than 20,000 inhabitants, while the rest live in smaller centers. Respondents live in different Italian regions, from north to south, but a large share of them are located in the northern region of Piedmont (approximately 60%).

When asked how they keep up with the news, the majority of the subjects declared that they often used online social media, while only about 17% used newspapers as frequently. Lastly, the great majority of the subjects (approximately 92%) were vaccinated or intended to get vaccinated.

After the treatment, every subject was asked their opinion about the people who did not want to get vaccinated, in terms of agreement on a 1-to-5 scale on the options listed in Table A1. Analogously, every subject was asked what they thought were the motivations of those who did not want to get vaccinated, as listed in Table A2. Lastly, with the same technique, we asked every subject what were their feelings about the evolution of the situation in Italy in the following 2 months and in the following 2 years (Table A3 and Table A4, respectively).

### 2.4. The Variables Not Analyzed

Among the questions present in the survey, some are not subject to our analysis, described below in Section 2.5. This is due to the fact that they were asked before the treatment took place or because the numbers involved are too small to make robust inference possible. In this section, we describe these questions, which are denoted in Figure 1 as question M (presented in two versions, one for vaccinated subjects and the other for non-vaccinated subjects) and question I. More detailed information about these variables are given in the Appendix A.

### 2.5. Empirical Strategy

The study’s four questions of interest were O1, O2, F1, and F2 (Figure 1), which were obtained after treatment. Each subject was given a list of several options on which to express their agreement on a Likert scale [42], as detailed in Table A1, Table A2, Table A3 and Table A4. To retain as much information as possible while reducing the number of variables observed, we applied categorical principal component analysis (PRINCALS, [43]) to each of the four questions. This technique allowed us to identify the component structure of students’ beliefs about the Italian COVID-19 vaccination campaign, with specific regard to ordinal response variables [44]. In other words, we could reduce each matrix of answers to the few uncorrelated and independent components that account for a large part of the variability observed in the data [45].

The main advantage of using categorical principal component analysis is in the implementation of the regression models and their interpretation. The approach allowed us to obtain a set of principal components that could be considered as dependent variables and could be regressed on the independent variables (see Table 1). The results of these regressions are presented in Table 1. More details about the original answers and the principal component analysis can be found in Appendix B.

After this elaboration, the original survey was transformed into 12 dependent variables (i.e., 3 principal components for each of the 4 questions of interest). These responses were used in separate linear regressions that depended on a set of individual characteristics and the four treatments. Specifically, we used the following regression equation:(1)$$y.PCi=\beta^{\prime }X+\gamma^{\prime }T+\epsilon ,$$
where y=a,b,c,d identifies the question, i=1,2,3 indicates the principal component, *X* is a matrix n×k for *n* respondents and *k* individual characteristics, and *T* is the n×4 treatments matrix. The treatments included two different kinds of arguments that aimed to orient adolescents’ opinions about vaccine hesitancy. The first argument highlighted the benefits of the vaccination campaign from the perspective of a public good or an individual good, while the second investigated the effects of information about mild and severe side effects of vaccines.

The principal components accounted for 62 to 80% of the total variance embedded in the question matrices. Given the relatively low number of participants in the sample, we reduced the number of individual characteristics used as covariates in the regressions to ensure the rejection of the null hypothesis in the F-test, which is that all the coefficients are jointly different from zero with $H_{0}:\beta_{1}=\ldots =\beta_{k}=\gamma_{1}=\ldots =\gamma_{4}=0$.

To obtain interpretable results, we performed different stepwise selections of variables for highlighting the best subset of predictors for each regression [46], sacrificing comparability across different regressions. Specifically, we selected the best subset that included the treatment variables. Specifically, we started with the full model that included all individual characteristics and treatments, then sequentially eliminated variables with the highest *p*-value until the rejection of the null hypothesis. The final subset of predictors was chosen mainly based on the Akaike information criterion (AIC), which is a measure of the relative quality of a statistical model for a given set of data, but also on the Schwarz and Bayesian criteria and on different $R^{2}s$. A more detailed description of the implemented stepwise variable selection is presented in Appendix C.

## 3. Results

In this section, we present the results of the regressions outlined in Section 2.5. We first examine the analysis of questions O1 and O2, which address opinions about anti-vaxxers and the beliefs underlying their motivations. It is important to note that these questions do not pertain to the motivations behind being pro-vaccine or anti-vaccine. Rather, they aim to capture participants’ beliefs regarding the motivations of anti-vaxxers, in other words, what participants think drives anti-vaxxers.

In the final subsection, we shift our focus to the analysis of questions F1 and F2, which investigate participants’ feelings about the short- and long-term evolution of the pandemic.

### 3.1. Opinions Concerning Anti-Vaxxers

After the treatment, the first question posed to students was O1, which asks “What do you think about those who do not want to get vaccinated?” This question elicits a personal judgment about individuals who were not willing to accept vaccination at the time. As described in Section 2.5, we employed principal component analysis as a dimensionality reduction method to identify a smaller number of variables that capture the most relevant information. The set of variables changes in each analysis.

As shown in Table A1, the first principal component, a.PC1, explains 39.48% of the total variance and is characterized by negative weights for each answer, particularly the one related to trust in science and vaccines. We refer to a.PC1 as *generic distrust*, to convey that it captures the notion that non-vaccinated individuals exhibit a general distrust of science and medicine, either due to selfishness or diffidence. Similarly, the second principal component, a.PC2 (VAF: 13.75%), is mainly influenced by the only two answers that specifically mention the word “vaccine,” and we label it *distrust of vaccines*.

Finally, the third principal component, a.PC3 (VAF: 11.55%), appears to reflect a positive judgment towards anti-vaxxers. This component captures the profile of an individual who values personal freedom, is not easily influenced by others, and exhibits courage. This may seem contradictory, but it is actually the profile of a *self-thinker*, someone who listens to others but ultimately reasons with their own mind and is brave enough to defend their ideas.

Table 2 displays the results of three regressions with the three principal components described above as dependent variables. Notably, stepwise selection suggests the same set of regressors for all three regressions, with the exception of a.PC2, which also includes a dichotomous variable indicating students from lyceums. Regarding the first regression, which corresponds to a.PC1, we observe that male students and those who are vaccinated or planning to get vaccinated tend to view non-vaccinated individuals as more distrustful of science and medicine. Conversely, low levels of education are associated with a lower tendency to attribute generic distrust to anti-vaxxers. In this case, none of the treatments display a significant impact on this opinion.

Regarding the regression for a.PC2, a negative relationship between a low-educated family background and pro-vaccination sentiment is evident, with the latter category being more likely to consider non-vaccinated individuals as specifically scared of vaccines. However, this is the only case that seems to be influenced by some of the treatments. Specifically, individuals in T3 and T4, who received information about adverse effects from vaccination, are more inclined to indicate vaccine hesitancy as the primary motivation for not getting vaccinated. Lastly, a.PC3 is negatively associated with male students, social media users, and vaccinated individuals. While the latter group’s disagreement with a positive view of anti-vaxxers is coherent, the fact that social media users do not hold favorable opinions of individuals who refuse vaccination is puzzling and could obscure mixed and confounding explanations, which we will discuss in more depth in Section 4.

Table 3 provides further evidence of the robustness of these results. In this case, participants were asked question O2, “What do you think was the motivation for those people who do not want to get vaccinated?” Thus, O2 captures a specific opinion on the motivations behind individuals who decline vaccination and can also be seen as a control for O1. As summarized in Table A2, the first component b.PC1 (VAF: 39.22%) reflects a *generic distrust* of science, medicine, and institutions as a whole, precisely mirroring a.PC1. The second component, b.PC2 (VAF: 13.81%), is primarily influenced by responses indicating hesitancy toward the COVID-19 vaccine and can be reasonably interpreted as skepticism towards this specific type of vaccine (*distrust of COVID-19 vaccine*). Finally, the third component, b.PC3 (VAF: 10.25%), is driven by a lack of perceived necessity for information on side effects, which could be interpreted as either an irreversible or irrational prejudice or as a strong trust. The fact that this component is also associated with beliefs that COVID-19 is not dangerous and with distrust of science and vaccines suggests that it could broadly signify a *denial* of the existence and impact of the COVID-19 pandemic.

Table 3 reveals a different set of covariates than those selected for O1 in Table 2. Notably, the variable for pro-vaxxers does not appear as relevant for attributing the motivation of *generic distrust* (b.PC1) in this case. The results for students from low-educated families are consistent with what was observed in a.PC1. We also observe a negative effect on *generic distrust* for those who occasionally use social networks. Additionally, we find an association with two of the proposed treatments. A positive relationship emerges for those who received information about free-riding, as in question O1. However, while T3 and T4 impacted the second component (a.PC2) in Table 2, they now impact the first component (b.PC1). For the second component, b.PC2, stepwise selection highlights the importance of acquiring information through both social channels and newspapers. It is worth noting that the coefficient of the variable *vaccinated: yes* is positive and significant for a.PC2, *distrust of vaccines*, while it is negative and significant for b.PC2, *distrust of COVID-19 vaccine*. Thus, we can hypothesize that pro-vaxxers believe that non-vaxxers distrust vaccines in general, and this distrust also applies to the COVID-19 vaccine.

Finally, the regression about *COVID-19 deniers* includes the most covariates. However, only two variables display significant effects: propensity to vaccination and lower levels of family education, both of which produced strong impacts in previous regressions. Specifically, we observe a contrasting effect between being favorable to vaccines (positive association) and coming from a low-educated family (negative association).

### 3.2. Feelings about the Evolution of the COVID-19 Pandemic

In this section, we analyze the second block of two questions asked after the treatment, F1 and F2, which concern the students’ feelings about the evolution of the COVID-19 pandemic in the short term and long term. The first question (F1) asks, “How do you think the pandemic situation will evolve in Italy in the next two months?” This question aims to capture participants’ impressions of the short-term evolution of the pandemic situation. As summarized in Table A3, the first principal component, c.PC1 (VAF: 39.27%), is mainly driven by three options pointing towards the idea that the virus will remain dangerous. Thus, we label it as *generic pessimism*. The second component, c.PC2 (VAF: 20.67%), is primarily driven by three factors indicating that COVID-19 will weaken itself and become less severe due to the growing adoption of vaccines. Accordingly, we label it as *optimism towards the COVID-19 vaccine*. Finally, the third component, c.PC3 (VAF: 14.84%), is mainly related to the belief that the virus will not be dangerous due to variants but because of too many people not getting vaccinated. We label this component *pessimism towards others’ behavior*.

Table 4 summarizes the results of the three regressions for question F1. Interestingly, students from lyceums are less likely to experience generic pessimism towards the short-term evolution of the pandemic situation (c.PC1). This belief is more common among those who do not receive information from social media or frequently read traditional newspapers. Among the four treatments, the one emphasizing the importance of vaccines as a public good (T1) significantly influences students’ feelings about the future of the pandemic, positively associating with generic pessimism. Moving to the second regression (c.PC2), students who are optimistic towards vaccines are more likely to come from highly educated families, not receive information on social networks, and not identify as binary gender. Surprisingly, those who are in favor of vaccines exhibit pessimism towards others’ behavior (c.PC3), unlike those who obtain information from social media.

The same options were proposed to investigate long-term feelings through question F2: “How do you think the pandemic situation will evolve in Italy in the next two years?” As with question F1, the first component d.PC1 (VAF: 41.34%) represents *generic distrust*, and the second component d.PC2 (VAF: 26.22%) is labeled *optimism towards the COVID-19 vaccine* since it is primarily driven by the belief that the virus will lose strength due to the adoption of vaccines. Conversely, the third component d.PC3 (VAF: 12.96%) is associated with a contrasting opinion, reflecting a sort of hesitation due to optimism towards natural virus weakening and pessimism towards the success of the vaccine campaign. It is classified as generic *uncertainty*.

Table 5 presents the results for the three regressions on the components described above. Similar to c.PC1, students from lyceums and those from lower-educated families are less likely to experience generic pessimism towards the evolution of the pandemic. Additionally, receiving information about vaccination as as a public good is positively associated with generic pessimism, consistent with c.PC1. Concerning d.PC2, optimism towards vaccines increases as parental education decreases. Although the coefficient for vaccinated students is barely non-significant, it is possible to assume that it is positively associated with d.PC2. To confirm this, we performed a one-sided t-test, which rejects the null hypothesis $H_{0}:\beta >0$ at a 90% confidence interval. Finally, vaccinated students are positively associated with *uncertainty* (d.PC3), as are those who received any of the treatments except the third.

## 4. Discussion

### 4.1. Key General Findings

The results of this pilot survey show that in Italy, about four months after the Italian government authorized the immunization campaign even for adolescents, a positive attitude toward vaccination can be found, versus only around 8% of the sample rejecting it. This is consistent with the data reported by the Italian government during the time of this study [47], and with the rates observed in previous studies [29,48]. In addition, it is also worth mentioning that vaccination for COVID-19 was not mandatory for this age group and that, at the time of the survey, individuals who wanted to get vaccinated possibly could not do it right away due to long waiting lists.

Our findings suggest that, in this current climate of widespread distrust and conspiracy theories, many students perceive others as exhibiting a general sense of skepticism towards both institutions and science. This pessimism extends not only to medicine and institutions, but also to virus mutations and people’s behavior in response to the pandemic. Once we disentangled this macro-effect, we found that our subjects focused their attention on vaccines, noting a lack of trust in the COVID-19 vaccine itself. This lack of trust may be due to the rapid response of the scientific community, which may have been perceived as disjointed from an efficient communication campaign. Despite this lack of trust, our subjects expressed general optimism about the efficacy of the vaccine, but were pessimistic about the response of the general population. It is worth noting that, after cleaning the data and controlling for the effect of vaccine distrust, a third component emerged regarding conspiracy theorists themselves. These individuals completely deny the existence and/or danger of the virus, and are viewed as brave critical-thinkers by their supporters. It is unlikely that much can be done to convince this small segment of the population, but it is crucial to focus on rebuilding trust among people, institutions, and the scientific community.

### 4.2. Findings about Individual Characteristics

Research involving medium to large samples has not found an association with socio-demographic variables [22,29,48], or if it has, it is that being female has been positively associated with a higher likelihood of being hesitant about the vaccine, as in a study conducted on this matter in China [49]. More studies would be needed to explore any influence of gender differences, mostly investigated in adults [50], and education type, on the intention to get vaccinated. All of these are factors that should be taken into account when developing effective and tailored communication strategies on such sensitive subjects as vaccination. Coherently, our study found no significant gender difference in adolescents’ perceptions of anti-vaxxer attitudes and their feelings about the evolution of the pandemic. However, our results suggest that these beliefs are more strongly correlated with environmental factors such as type of education and family background, at least during this stage of life. Nonetheless, females were more likely to view anti-vaxxers as self-thinkers. While it is difficult to explain this finding, we can rule out the presence of selection bias, as females comprised roughly two-thirds of both pro- and anti-vaxxer respondents. Further research would be needed to better understand this result, which could be due to factors such as gender socialization [51] or perceived agency [52].

Interestingly, we found a negative correlation between the intense usage of social media and thinking that anti-vaxxers are free thinkers or brave people. Since a positive relationship has been shown in the literature between the use of social media and being a vaccine rejecter [53], we may assume that probably our findings are due to the parental influence on vaccination choices, regardless of the information received from social media by adolescents, supporting the development and exercise of their critical thinking.

We also discovered that both vaccine accepters and rejecters consider those between the ages of 35 and 65 to be the most irresponsible towards coping with COVID-19, which paradoxically is the age group of their parents. These beliefs could be related to several perceptions, for example, that people, once vaccinated, behave in an unsafe way (e.g., not wearing a mask), or that by getting vaccinated they have exposed themselves to the side effects of the vaccine. These matters were widespread during the pandemic and once again they highlight the moral duty to develop transparent communication strategies, as free and conscious choices cannot be achieved if one fails to provide adequate information [54,55].

Lastly, concerning familial background, our study found that adolescents with lower levels of family education tend to be more naive and indulgent towards COVID-19 deniers, potentially making them more susceptible to anti-vaccine messaging. Our study found that these adolescents tend to attribute lower levels of generic distrust to both anti-vaxxers and vaccines compared to other study participants. However, they are not associated with a positive opinion of anti-vaxxers as independent thinkers. This suggests that these adolescents may be more malleable and receptive to information and awareness campaigns targeting this population group.

### 4.3. Opinions about Anti-Vaxxers’ Motivations

Nonetheless, the main goal of this study was to identify adolescents’ opinions regarding those who refuse to get vaccinated, and their related motivations. Regardless of whether they were accepters or rejecters of COVID-19 vaccines, the largest part of the variance among our participants’ answers about their beliefs toward those who do not get vaccinated (question O1 and O2) relate to what we identified as a generic distrust. In other words, a lack of trust transversal among science, medicine, and institutions. Only secondarily did their beliefs focus specifically on vaccines and/or the COVID-19 vaccine in particular. Furthermore, having been exposed to information about the vaccine’s adverse effects (treatments T3 and T4) makes them more likely to endorse this distrust of vaccines. This result suggests that adolescents seem sensitive even to mild pieces of information about vaccines’ adverse effects. Namely, information about vaccines is more effective in influencing their opinion towards anti-vaxxers, moving it against vaccines, rather than generic distrust. From this point of view, our findings suggest that communication strategies should also be devised with the assistance of psychologists, in order to cope with widespread fear and thus achieve greater vaccination coverage.

We show also that the idea of vaccine rejecters as self-thinkers is almost always negatively associated with the included characteristics. Apparently, only anti-vaxxers aged between 16 and 17 consider them as brave people, able to bear the weight of uncertainty and fear and able to listen to several conflicting voices, then elaborating them in a consistent critical opinion.

In general, vaccine rejecters are perceived as people who have not developed a good level of trust in official sources on health matters. These findings are in line with previous studies showing an association between trust relationships, including trust in healthcare professionals, the health system, the government, and friends and family members and vaccination uptake [56,57]. In this regard, strategies aimed at achieving a high percentage of vaccinated people, even among adolescents, and ending a pandemic must necessarily aim on the one hand to build and maintain trust in public health institutions, their messages and the science upon which their communication is based (see page 13 in [54]), and on the other, to develop the virtue of resilience toward information that negatively affects trust, by investing in reflective critical thinking skills, and also towards self-knowledge [58].

### 4.4. Opinions about the Evolution of the COVID-19 Pandemic

A second part of our results concerns the opinions and perceptions about the evolution of the pandemic in the short and long term. Specifically, the largest part of the variance accounted for relates to a short-term pessimism, mainly related both to the emergence of new COVID-19 variants and to distrust in people’s adherence to the vaccination campaign. This pessimism is not strongly linked to the use of social media and, analogously, not strongly associated with the usage of newspapers. Regarding the long term, the situation is also similar, although the lower the familial background education (and/or being enrolled into a lyceum), the lower this generic pessimism. Nonetheless, a high usage of social media is associated with a lower optimism toward vaccines and a lower pessimism toward others’ behavior. Our hypothesis is that, although social media contribute to spread fear against vaccines’ efficacy, they are not able to convince adolescents that anti-vaxxers are anything but a noisy minority. These results highlight the relevance of parental education in affecting vaccination opinions, compared to the social media influence. This suggests that teenagers’ hesitation and rejection of vaccines is also an ethical concern and, therefore, solutions cannot be sought outside this front.

In our study, the subjects seemed to be particularly sensitive to considering the vaccine as a public good (treatment T1). From this point of view, previous studies [34,35] have found a beneficial effect of media encouragement on vaccination intentions and the idea that the vaccine is a common good. Nonetheless, this effect seems to be mediated not only by personal attitudes, but also by injunctive social norms, which push people towards modifying their expectations about others’ behavior. Coherently, in a period in which the voices of anti-vaxxers seem relatively strong, our attempt at stimulating our subjects to considering vaccines as public goods (via T1) has made them more pessimistic and more uncertain about the response of the whole society.

### 4.5. Limitations of This Study and Further Research

This study presents an opportunity for reflection on future tailored vaccination strategies aimed at meeting adolescents’ concerns. However, it also comes with limitations that may affect the generalizability and sustainability of the findings. For instance, a small sample size and reliance on self-reported data may limit the study’s ability to capture the complexity of vaccine hesitancy, which could also be influenced by factors such as access to accurate information and healthcare infrastructure. Moreover, the study’s focus on Italian adolescents’ perceptions of vaccine hesitants’ motivations may not fully capture the attitudes and beliefs about vaccination across different cultural, social, and political contexts.

It is important to note, that the findings of this study may not be directly generalizable to other countries or cultures due to differences in attitudes, beliefs, and contextual factors that could affect vaccine hesitancy. Nonetheless, globalization has increased the interconnectedness of people and cultures across the world, making the study’s focus on adolescents’ attitudes about vaccination and social media influence relevant to other contexts. Therefore, further research is necessary to understand the nuanced differences in vaccine hesitancy across various contexts and to develop tailored public health strategies that consider the specific needs and circumstances of different populations.

Despite these limitations, the findings suggest that adolescents with lower levels of family education tend to be more susceptible to anti-vaccine messaging, indicating the need for tailored public health strategies that consider the specific needs and circumstances of different populations. To overcome some of the limitations, future research could explore additional questions about parental compulsion or familial obligation for getting or not getting vaccinated. Additionally, due to the relatively small fraction of the Italian adolescent population that is not vaccinated, obtaining a large enough sample size of non-vaccinated individuals will be crucial for making robust statistical analyses about these individuals.

## 5. Conclusions

The results of this study may be useful in further shedding light upon the fact that the propensity not to adhere to the vaccination campaign is a complex phenomenon that should be studied in a multi-parametric approach, in relation to age groups. In this regard, it might be interesting to also involve adolescents in the development of communication strategies together with experts in this field and psychologists, to get targeted inputs, understand risk behavior, and develop prevention paths.

Overall, our study suggests that Italian adolescents are particularly sensitive to the topic of vaccines’ adverse effects when asked to speculate about anti-vaxxers’ beliefs. This finding underscores the importance of transparency in vaccination campaigns, both in Italy and globally. It is crucial to communicate clearly about the scientific method and potential adverse effects of vaccines, while avoiding excessive magnification of any concerns. An excessive negation on the potential risks of a new vaccine developed in record time could inadvertently bolster the arguments of deniers and conspiracy theorists. Instead, a balanced approach that acknowledges potential risks while emphasizing the benefits of vaccination is essential for promoting public trust and increasing vaccination rates. On the other hand, our study suggests that information about vaccines as public and private goods can influence students’ perceptions of the pandemic’s evolution.

Communication campaigns should aim to fully explain the benefits of widespread vaccination, not only for individual health but also for collective and social well-being. These campaigns should be targeted at younger individuals from lower socioeconomic backgrounds and less urbanized areas, who may be more skeptical but also more receptive to persuasion at this stage of life. By addressing their concerns and emphasizing the importance of participation in vaccination programs, we can increase overall vaccination rates and reduce the impact of the pandemic.

In other words, preparedness for future pandemic events requires the development of social compliance with health recommendations based on reflective critical thinking and on building a grounded trust on official sources of information, both on the side of teenagers and their parents. According to the hints obtained through our treatments, we believe that promoting reflective critical thinking in this context requires reinforcing the benefits of vaccines as public goods and increasing transparency regarding the scientific methods used in the development and distribution of vaccines worldwide. The significant portion of the variance in our results capturing a generic distrust may indicate a lack of trust in community leaders and their ability to guide the population during times of crisis. Lastly, the fact that adolescents consider adults as the most irresponsible actors in the pandemic crisis, highlights the need for significant efforts to bridge the generational gap and rebuild trust between different age groups.

Overall, our study emphasizes the importance of targeted communication and would promote collaboration between public health authorities, educational institutions, and social media platforms aimed at pandemic preparedness and improvements in public health outcomes.

## Figures and Tables

**Figure 1 vaccines-11-00967-f001:**
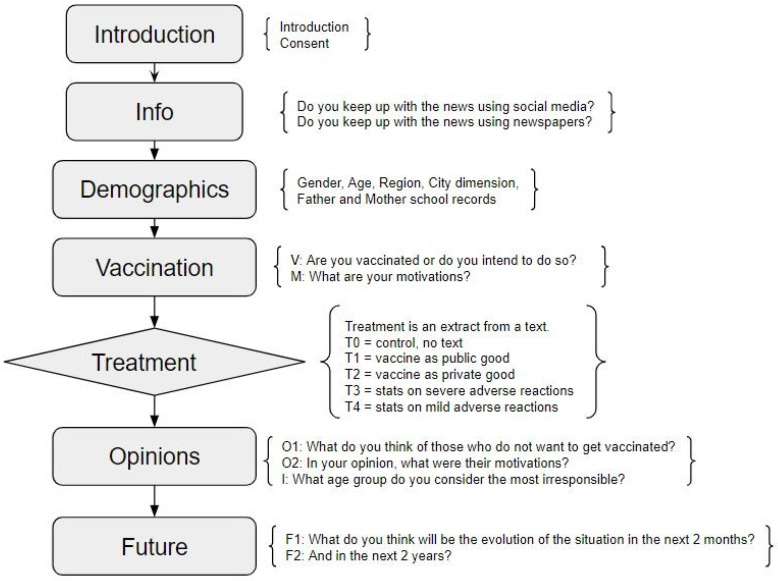
Survey flow.

**Table 1 vaccines-11-00967-t001:** Descriptive statistics of the variables used in the analysis. For clarity, throughout the paper, categories pertaining to the same variable are grouped by color.

	Full Sample	T0	T1	T2	T3	T4
	(%)	(%)	(%)	(%)	(%)	(%)
**Independent variables**						
Gender: Females	65.82	66.67	58.54	73.33	67.11	64.20
Gender: Males	27.34	22.22	36.59	24.00	26.32	27.16
Gender: Other	6.84	11.11	4.88	2.67	6.58	8.64
Age: 12–15	23.80	24.69	26.83	20.00	23.68	23.46
Age: 16–17	48.86	45.68	50.00	50.67	47.37	50.62
Age: 18–19	27.34	29.63	23.17	29.33	28.95	25.93
Mother’s ed.: Uni	25.57	17.28	34.15	26.67	28.95	20.99
Mother’s ed.: HS	39.75	43.21	35.37	45.33	38.16	37.04
Mother’s ed.: No HS	34.68	39.51	30.49	28.00	32.89	41.98
No lyceum HS	26.08	25.93	24.39	24.00	30.26	25.93
Lyceum (ref: other HS)	73.92	74.07	75.61	76.00	69.74	74.07
Big city (>20,000 inhabitants)	65.57	59.26	65.85	68.00	65.79	69.14
Small city (<20,000 inhabitants)	34.43	40.74	34.15	32.00	34.21	30.86
North/center	85.32	87.65	84.15	88.00	80.26	86.42
South/islands	14.68	12.35	15.85	12.00	19.74	13.58
Social: Rarely	4.30	3.70	7.32	2.67	1.32	6.17
Social: Sometimes	25.06	22.22	26.83	25.33	21.05	29.63
Social: Often	70.63	74.07	65.85	72.00	77.63	64.20
News: Rarely	40.00	32.10	43.90	33.33	47.37	43.21
News: Sometimes	42.53	49.38	42.68	48.00	35.53	37.04
News: Often	17.47	18.52	13.41	18.67	17.11	19.75
Vaccinated: No	7.59	7.41	7.32	2.67	13.16	7.41
Vaccinated: Yes	92.41	92.59	92.68	97.33	86.84	92.60
T0: Control group	20.51	-	-	-	-	-
T1: Public good	20.76	-	-	-	-	-
T2: Private good	18.99	-	-	-	-	-
T3: Severe adverse effects	19.24	-	-	-	-	-
T4: Mild adverse effects	20.51	-	-	-	-	-
	**Mean**	**Mean**	**Mean**	**Mean**	**Mean**	**Mean**
**Dependent variables**						
a.PC1	0.000	0.101	−0.253	0.054	0.031	0.076
a.PC2	0.000	0.104	−0.192	−0.061	0.007	0.173
a.PC3	0.000	0.223	−0.035	−0.089	−0.019	−0.087
b.PC1	0.000	−0.196	0.048	−0.079	0.157	0.073
b.PC2	0.000	−0.072	0.192	0.061	−0.007	−0.173
b.PC3	0.000	−0.054	0.108	0.017	−0.125	0.047
c.PC1	0.000	−0.096	0.217	−0.038	−0.001	−0.088
c.PC2	0.000	−0.016	0.005	−0.148	0.013	0.135
c.PC3	0.000	−0.036	−0.160	0.119	0.067	0.026
d.PC1	0.000	−0.101	0.253	−0.054	−0.031	−0.076
d.PC2	0.000	0.111	−0.006	−0.026	−0.114	0.027
d.PC3	0.000	−0.223	0.035	0.089	0.019	0.087
No. observations	395	81	82	75	76	81

**Table 2 vaccines-11-00967-t002:** Motivations of anti-vaxxers. Regressions as modeled in Equation (Equation 1). The dependent variables are the three main principal components, a.PC1, a.PC2, and a.PC3, described in Table A1, corresponding to question O1: “What do you think about those who do not want to get vaccinated?”.

	a.PC1: Generic Distrust			a.PC2: Distrust of Vaccines			a.PC3: Self-Thinkers		
	Coeff.	St. Dev.	*p*-Value	Coeff.	St. Dev.	*p*-Value	Coeff.	St. Dev.	*p*-Value
Gender: Males	0.259	0.111	*0.020*	0.004	0.115	0.973	−0.239	0.115	*0.038*
Gender: Other	−0.143	0.195	0.466	−0.062	0.203	0.759	−0.115	0.202	0.569
Age: 16–17	−0.186	0.122	0.129	0.116	0.126	0.358	0.280	0.126	*0.026*
Age: 18–19	−0.012	0.139	0.929	0.119	0.144	0.409	0.178	0.144	0.217
Mother’s ed.: HS	−0.272	0.124	*0.029*	−0.180	0.129	0.163	−0.091	0.128	0.480
Mother’s ed.: No HS	−0.423	0.130	*0.001*	−0.395	0.136	*0.004*	−0.209	0.134	0.119
Lyceum (ref: other HS)	-	-	-	−0.041	0.118	0.731	-	-	-
Social: Sometimes	−0.133	0.336	0.600	−0.372	0.263	0.158	−0.055	0.261	*0.036*
Social: Often	0.101	0.243	0.677	−0.255	0.252	0.312	−0.443	0.251	*0.078*
Vaccinated: Yes	0.698	0.185	*0.000*	0.357	0.192	*0.064*	−0.502	0.191	*0.009*
T1: Public good	0.132	0.153	0.388	0.230	0.158	0.146	−0.019	0.157	0.905
T2: Private good	0.112	0.155	0.471	0.072	0.161	0.652	0.108	0.160	0.500
T3: Severe adverse effects	0.220	0.154	0.155	0.308	0.160	*0.055*	−0.094	0.159	0.557
T4: Mild adverse effects	0.035	0.151	0.817	0.319	0.156	*0.042*	0.171	0.156	0.273
Intercept	−0.493	0.336	0.142	−0.091	0.353	0.798	0.876	0.346	*0.012*
No. obs.	395			395			395		
R-squared	0.117			0.057			0.059		
Adj. R-squared	0.087			0.022			0.027		
F-test	3.879 ***			1.632 *			1.834 **		

To improve readability, the *p*-values in italics correspond to coefficients that are significant at a 10% level. Significance level of F-test: * 10%, ** 5%, *** 1%.

**Table 3 vaccines-11-00967-t003:** Motivations of anti-vaxxers. Regressions as modeled in Equation (Equation 1). The dependent variables are the three main principal components, b.PC1, b.PC2, and b.PC3, described in Table A2, corresponding to question O2: “What do you think was the motivation for those people who do not want to get vaccinated?”.

	b.PC1: Generic Distrust			b.PC2: Distrust of COVID-19 Vaccine			b.PC3: COVID-19 Deniers		
	Coeff.	St. Dev.	*p*-Value	Coeff.	St. Dev.	*p*-Value	Coeff.	St. Dev.	*p*-Value
Gender: Males	-	-	-	-	-	-	0.064	0.114	0.579
Gender: Other	-	-	-	-	-	-	−0.108	0.201	0.591
Age: 16–17	−0.026	0.126	0.834	-	-	-	0.193	0.125	0.123
Age: 18–19	0.148	0.143	0.302	-	-	-	0.056	0.143	0.696
Mother’s ed.: HS	−0.102	0.128	0.428	-	-	-	−0.054	0.128	0.671
Mother’s ed.: No HS	−0.337	0.132	*0.011*	-	-	-	−0.292	0.135	*0.031*
Lyceum (ref: other HS)	-	-	-	-	-	-	0.153	0.117	0.190
Social: Sometimes	−0.430	0.261	*0.099*	−0.432	0.259	*0.059*	−0.175	0.261	0.503
Social: Often	−0.229	0.250	0.361	−0.264	0.247	0.285	0.147	0.250	0.556
News: Sometimes	-	-	-	−0.115	0.110	0.296	-	-	-
News: Often	-	-	-	−0.299	0.142	*0.036*	-	-	-
Vaccinated: Yes	-	-	-	−0.661	0.189	*0.001*	0.672	0.191	*0.000*
T1: Public good	0.217	0.157	0.166	0.055	0.154	0.720	0.104	0.156	0.507
T2: Private good	0.090	0.159	0.572	−0.093	0.157	0.555	−0.014	0.159	0.931
T3: Severe adverse effects	0.329	0.159	*0.039*	0.163	0.158	0.303	−0.055	0.158	0.730
T4: Mild adverse effects	0.287	0.156	*0.067*	0.177	0.155	0.254	0.090	0.155	0.561
Intercept	0.215	0.286	0.454	0.946	0.329	*0.004*	−0.610	0.350	*0.082*
No. obs.	395			395			395		
R-squared	0.050			0.065			0.073		
Adj. R-squared	0.025			0.043			0.039		
F-test	2.006 **			2.978 ***			2.138 ***		

To improve readability, the *p*-values in italics correspond to coefficients that are significant at a 10% level. Significance level of F-test: ** 5%, *** 1%.

**Table 4 vaccines-11-00967-t004:** Feelings about the short-term evolution of the COVID-19 pandemic. Regressions as modeled in Equation (Equation 1). The dependent variables are the three main principal components, c.PC1, c.PC2, and c.PC3, described in Table A3, corresponding to question F1: “How do you think the pandemic situation will evolve in Italy in the next two months?”.

	c.PC1: Generic Pessimism			c.PC2: Optimism toward Vaccines			c.PC3: Pessimism toward Others’ Behavior		
	Coeff.	St. Dev.	*p*-Value	Coeff.	St. Dev.	*p*-Value	Coeff.	St. Dev.	*p*-Value
Gender: Males	-	-	-	0.096	0.114	0.402	0.153	0.115	0.182
Gender: Other	-	-	-	0.398	0.201	*0.048*	0.050	0.204	0.808
Mother’s ed.: HS	-	-	-	−0.248	0.127	*0.051*	-	-	-
Mother’s ed.: No HS	-	-	-	−0.129	0.131	0.327	-	-	-
Lyceum (ref: other HS)	−0.280	0.115	*0.015*	-	-	-	0.162	0.115	0.161
Small City (<20,000 inhabitants)	-	-	-	0.177	0.106	*0.095*	−0.126	0.107	0.241
Social: Sometimes	−0.530	0.261	*0.043*	−0.724	0.259	*0.005*	−0.503	0.264	*0.057*
Social: Often	−0.406	0.249	0.103	−0.581	0.248	*0.020*	−0.447	0.252	*0.077*
News: Sometimes	0.052	0.110	0.636	-	-	-	-	-	-
News: Often	0.341	0.144	*0.035*	-	-	-	-	-	-
Vaccinated: Yes	-	-	-	−0.581	0.248	*0.020*	0.412	0.192	*0.032*
T1: Public good	0.329	0.155	*0.035*	−0.001	0.156	0.993	−0.168	0.157	0.287
T2: Private good	0.071	0.158	0.651	−0.060	0.159	0.707	0.129	0.161	0.421
T3: Severe adverse effects	0.103	0.158	0.516	0.017	0.158	0.915	0.131	0.159	0.411
T4: Mild adverse effects	0.009	0.156	0.951	0.157	0.155	0.313	0.036	0.157	0.817
Intercept	0.441	0.278	0.113	1.060	0.349	*0.003*	−0.083	0.340	0.806
No. obs.	395			395			395		
R-squared	0.054			0.069			0.046		
Adj. R-squared	0.032			0.040			0.018		
F-test	2.442 **			2.366 ***			1.664 *		

To improve readability, the *p*-values in italics correspond to coefficients that are significant at a 10% level. Significance level of F-test: * 10%, ** 5%, *** 1%.

**Table 5 vaccines-11-00967-t005:** Feelings about the long-term evolution of the COVID-19 pandemic. Regressions as modeled in Equation (Equation 1). The dependent variables are the three main principal components, d.PC1, d.PC2 and d.PC3, described in Table A4, corresponding to question F2: “How do you think the pandemic situation will evolve in Italy in the next two years?”.

	d.PC1: Generic Pessimism			d.PC2: Optimism toward COVID-19 Vaccine			d.PC3: Uncertainty (Virus vs. Vaccine Campaign)		
	Coeff.	St. Dev.	*p*-Value	Coeff.	St. Dev.	*p*-Value	Coeff.	St. Dev.	*p*-Value
Mother’s ed.: HS	−0.375	0.126	*0.003*	0.267	0.130	*0.041*	−0.043	0.130	0.742
Mother’s ed.: No HS	−0.336	0.132	*0.011*	0.297	0.135	*0.028*	0.138	0.135	0.307
Lyceum (ref: other HS)	−0.294	0.114	*0.011*	-	-	-	-	-	-
Small City (<20,000 inhabitants)	−0.152	0.104	0.146	-	-	-	-	-	-
South/islands (ref: north/center)	-	-	-	-	-	-	0.199	0.146	0.176
News: Sometimes	-	-	-	0.142	0.113	0.210	-	-	-
News: Often	-	-	-	0.095	0.148	0.521	-	-	-
Vaccinated: Yes	-	-	-	0.277	0.192	0.151	0.438	0.191	*0.022*
T1: Public good	0.289	0.155	*0.063*	−0.056	0.159	0.724	0.259	0.157	*0.022*
T2: Private good	0.009	0.158	0.953	−0.121	0.161	0.454	0.308	0.160	*0.055*
T3: Severe adverse effects	0.006	0.157	0.969	−0.156	0.162	0.336	0.259	0.160	0.105
T4: Mild adverse effects	−0.005	0.155	0.974	−0.059	0.158	0.707	0.301	0.156	*0.054*
Intercept	0.473	0.178	*0.008*	−0.465	0.249	*0.062*	−0.083	0.340	0.806
No. obs.	395			395			395		
R-squared	0.060			0.027			0.037		
Adj. R-squared	0.040			0.004			0.017		
F-test	3.074 ***			1.181			1.872 *		

To improve readability, the *p*-values in italics correspond to coefficients that are significant at a 10% level. Significance level of F-test: * 10%, *** 1%.

## Data Availability

The data that support the findings of this study are available upon request from the corresponding author, AM.

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
