# Peer review of "Adolescents’ Opinions on COVID-19 Vaccine Hesitancy: Hints toward Enhancing Pandemic Preparedness in the Future"

_vaccines, 2023, doi:10.3390/vaccines11050967_

Round 1
Reviewer 1 Report
It is not clear how and to all intents and purposes the opinions and perceptions of adolescents on the evolution of the Covid-19 pandemic influence enhancing pandemic preparedness.
Author Response
We thank the Reviewer for their comments. Please, see the uploaded file for a detailed reply.

Reviewer 2 Report
I have the following comments
1. Introduction in not focussed and must be shortened
2. Objective of the study is not very clearly stated.
3. Manuscript is very difficult to read and understand
4. 27% o fteh participants were of 18 plus age group. They can nor be consdired as adolescent. They shall be excluded froma nalysis.
Author Response

(The authors gave the same response as above.)

Reviewer 3 Report
The paper entitled "Adolescents’ opinions on COVID-19 vaccine hesitancy: hints toward enhancing pandemic preparedness in the future" aims to assess vaccine hesitancy in Italian adolescents by exploring their opinions and perceptions towards vaccination. The study was conducted using a randomized survey experiment on a sample of 395 high school students in different regions of Italy. The results show that vaccinated individuals tend to have a higher level of generic distrust in science towards anti-vaccine behavior, while excluding distrust of COVID-19 vaccines. The mother's education was identified as the most influential factor in vaccine hesitancy among adolescents.
The paper provides valuable insights into the perceptions and opinions of Italian adolescents towards vaccination and highlights the need for tailored strategies to promote vaccination coverage among this population. The study's findings suggest that a lack of education among parents contributes to vaccine hesitancy among adolescents. Therefore, the authors recommend developing social compliance with health recommendations based on reflective-critical thinking and building trust in official sources of information for both teenagers and their parents.
However, the paper needs some improvements before publication. The introduction should provide a more comprehensive background of the study and its potential applications, while the literature review should be updated and incorporate recent research related to the topic. Therefore, the authors should consider incorporating more recent references to provide a comprehensive review of the current state of research on vaccine hesitancy among adolescents.
Why is it important to study vaccine hesitancy among adolescents?
In what ways can the authors enhance the introduction and literature review of their paper?
What motivates Italian adolescents to make anti-vaccine decisions?
How did the authors conduct their survey experiment on Italian high school students in a randomized manner?
What were the key findings of the study regarding vaccine hesitancy among Italian adolescents?
To what extent does gender impact vaccine hesitancy among Italian adolescents?
How does parental education level influence vaccine hesitancy among Italian adolescents?
What specific strategies can be implemented to increase vaccination coverage among Italian adolescents?
How can reflective-critical thinking be incorporated to foster social compliance with health recommendations among Italian adolescents?
What can be done to establish trust in official sources of information among Italian adolescents and their parents?
What are the implications of the study's results for future pandemic preparedness efforts?
To what extent can the study's findings be generalized to other countries?
What are some of the possible limitations of the study?
What ethical considerations should be taken into account when conducting a survey experiment on adolescents?
What implications do the study's findings have for vaccine policy in Italy?
How can the study's results contribute to the global effort to increase vaccination coverage?
What role does science communication play in addressing vaccine hesitancy among adolescents?
How can public health campaigns use the study's findings to encourage vaccination among Italian adolescents?
What impact could vaccine hesitancy among adolescents have on herd immunity?
Author Response

(The authors gave the same response as above.)

Round 2
Reviewer 2 Report
Thanks fo rthe responses
Reviewer 3 Report
After thorough review, your revisions have made the manuscript suitable for acceptance.